# Complex effects of kinase localization revealed by compartment-specific regulation of protein kinase A activity

**Rebecca LaCroix[1,2], Benjamin Lin[1,2,3], Tae-Yun Kang[1,2], Andre Levchenko[1,2]***

[1]Department of Biomedical Engineering, Yale University, New Haven, United States; [2]Yale Systems Biology Institute, Yale University, West Haven, United States; [3]Department of Cell Biology, Skirball Institute of Biomolecular Medicine, NYU Langone Health, New York, United States

**Abstract** Kinase activity in signaling networks frequently depends on regulatory subunits that can both inhibit activity by interacting with the catalytic subunits and target the kinase to distinct molecular partners and subcellular compartments. Here, using a new synthetic molecular interaction system, we show that translocation of a regulatory subunit of the protein kinase A (PKA-R) to the plasma membrane has a paradoxical effect on the membrane kinase activity. It can both enhance it at lower translocation levels, even in the absence of signaling inputs, and inhibit it at higher translocation levels, suggesting its role as a linker that can both couple and decouple signaling processes in a concentration-dependent manner. We further demonstrate that superposition of gradients of PKA-R abundance across single cells can control the directionality of cell migration, reversing it at high enough input levels. Thus, complex in vivo patterns of PKA-R localization can drive complex phenotypes, including cell migration.

**\*For correspondence:**
andre.levchenko@yale.edu

**Competing interest:** The authors declare that no competing interests exist.

## Editor's evaluation

This is a very thorough and important study demonstrating quantitative control of signaling through changes in the abundance and localization of a regulatory kinase subunit. The authors use live imaging experiments in microfluidic devices to reveal nonmonotonic dependence of PKA activity on the level of its regulatory subunit and provide evidence that it translates into corresponding changes of cell polarization and cell migration. Moreover, they provide a mathematical model that explains the underlying mechanism.

## Introduction

In intracellular signal transduction, the information is encoded in molecular interactions involving recognition of diverse substrates by specific enzymes. These interactions are facilitated by large sets of adapter and scaffold proteins linking the activated enzymes to substrates within specific subcellular compartments, controlling dynamic modifications of protein localization and local concentrations of effector molecules (*Langeberg and Scott, 2015*). Furthermore, the enzymes, such as the diverse and abundant kinases involved in cell signaling and other functions, frequently contain covalently linked regulatory and catalytic subunits, with the regulatory subunits controlling both the activity of the enzyme and its interactions with other enzymes and substrates. However, the enzymes belonging to the family of protein kinase A (PKA) serine-threonine kinases do not follow this covalent linkage rule. Instead, a PKA molecule is a complex of two catalytic subunits (PKA-C) and a regulatory subunit dimer (PKA-R) that can dissociate following binding of two cyclic AMP (cAMP) molecules to

each of the PKA-R subunits (*Taylor et al., 2013*). This intricate organization of the kinase complex is further complicated by dynamically shifting subcellular pools of cAMP, tethering of the kinase and its substrates to a large family of differentially localized A-kinase anchoring proteins (AKAPs), localized phosphatase and phosphodiesterase activity, and the intrinsic inhibitory function of PKA-R (*Baillie, 2009*; *Wong and Scott, 2004*; *Zhang et al., 2012*). What emerges is a picture of structural and functional complexity of PKA signaling that is still incompletely understood in spite of decades of research and analysis.

Given the complexity of intermolecular interactions, the stoichiometry of multi-molecular complexes can have profound effects on the outcome of signaling processes. A particularly striking effect is observed if a linker protein, such as a scaffold molecule, can vary in its abundance. It has been shown, for example, for the MAPK signaling pathways, that a scaffold protein can enhance the pathway activity at an optimal level but can also inhibit it if its concentration exceeds the optimum (*Chapman and Asthagiri, 2009*; *Levchenko et al., 2000*). This 'combinatorial inhibition' effect (*Good et al., 2011*) suggests that variation of the relative abundance of the pathway components can modulate the pathway activity even if the input levels do not change. Given the structural complexity of PKA signaling, it is not clear whether and how the relative abundance of various signaling pathway components in different subcellular compartments may modulate the signaling outcomes. More specifically, it is not clear if PKA-R subunits would serve purely as inhibitors of PKA signaling (as expected due to their intrinsic inhibitory role, relieved only in the presence of high cAMP concentrations) or would potentially elevate the kinase activity by enhancing localization of the PKA holoenzyme to subcellular areas with increased signaling inputs.

Compartmentalized PKA signaling is important in a number of contexts including glucose homeostasis, cardiomyocyte contractility, cell cycle regulation, and cell migration (*Howe, 2004*; *Langeberg and Scott, 2005*; *Mauban et al., 2009*; *Wong and Scott, 2004*). In regulating cell migration, PKA is known to positively affect the activity of some cytoskeletal regulators (e.g., Rac1, Cdc42, α4β1 integrin) and negatively affect others (e.g., RhoA). Additionally, both inhibition and activation of PKA have been shown to have inhibitory effects on cell migration (*Howe, 2004*). As a result, PKA's role in regulation of cell migration is still unclear. Several studies utilizing FRET biosensors have identified gradients of PKA activity in migrating cells, with relatively high PKA activity at the cell front and relatively low activity at the rear, suggesting that spatial control of the kinase is involved in this process (*Lim et al., 2008*; *Paulucci-Holthauzen et al., 2009*; *Tkachenko et al., 2011*). Furthermore, the regulatory but not the catalytic subunit has been shown to be enriched in the pseudopods of cells in culture, suggesting that regulatory subunit localization may play a role in the regulation of cell migration (*Howe et al., 2005*).

Since PKA-R mediates anchoring of PKA to diverse intracellular locations, in large part due to its interactions with AKAPs, it is particularly important to explore whether and how its abundance at specific subcellular locations modulates the output of signaling activity both under the basal conditions and in response to specific stimulation. This analysis can benefit from a tool that can permit acute localization of PKA-R to a predefined cell location in the absence of direct pathway stimulation. We developed such a tool based on chemically inducible dimerization with the small, cell-permeable molecule rapamycin (*Banaszynski et al., 2005*). Rapamycin induces dimerization of two small, intracellularly transduced molecular components: FK506-binding protein (FKBP) and the FKPB-rapamycin domain (FRB), which can be tethered to proteins of interest as well as specific subcellular compartments. This technique has been used to study biochemical activity of different proteins (*Chu et al., 2014*; *Dagliyan et al., 2017*; *Dagliyan et al., 2013*; *Inoue et al., 2005*; *Karginov et al., 2010*). Previously, we demonstrated that this dimerization strategy, when combined with the use of a microfluidic device controlling spatial rapamycin distribution, can be used to study biomolecular systems controlling cell polarity and migration (*Lin et al., 2012*). For the current study, we have tethered FKBP to fluorescently labeled PKA-R, while anchoring FRB to the plasma membrane (PM), to enable dynamically controlled localization of PKA-R to the PM in a rapamycin-dependent fashion. Using this tool, we find that PKA-R can have unexpectedly complex regulatory effects on the activity and function of PKA at the PM, elucidating the potential role of PM PKA-R localization in normal and pathological cell function.

## Results

### Design and characterization of an inducible PKA-R translocation system

Our aim was to develop a synthetic tool to control PKA-R abundance in a specific subcellular location, with the ability to assess the real-time local subunit abundance. We found that of the four PKA-R isoforms (PKAR-Iα/β, PKAR-IIα, and PKAR-IIβ) in a standard HeLa cell line, the expression of PKAR-IIβ was particularly low (*Figure 1—figure supplement 1A* and D, rightmost lane), prompting us to use this isoform as the basis for development of the chemical dimerization-based tool for acutely controlling local PKA-R abundance. Specifically, to induce rapid and spatially controlled translocation of PKA-R to the PM, we utilized a rapamycin-based dimerization strategy combined with in-dish or in-chip control of rapamycin concentration similar to the strategy we previously used to control Rac1 function (*Lin et al., 2012*). We linked one binding partner of rapamycin, FKBP, to a fluorescently tagged PKA-RIIβ (PKAR-FKBP-FP) and the other, FRB, to the membrane-targeting sequence of Lyn kinase (Lyn$_{11}$-FRB, also referred to as Lyn-FRB throughout the text) (*Figure 1A*). Two color variants of the PKA-R construct were created to facilitate co-imaging with fluorescent biosensors and dyes (*Figure 1B* and *Figure 1—figure supplement 2*). In addition to our transient expression vectors, lentiviral Gateway expression vectors for PKAR-FKBP-FP and Lyn-FRB were created and integrated into the genome of HeLa cells for ease of experimentation. We will refer to these cell lines as HeLa PFM (mCherry variant) and HeLa PFY (YFP variant), respectively. Although, expectedly, transfection of the modified PKAR-IIβ changed the expression level of this subunit in both transiently transfected (to a lower degree) cells and stably transfected clones (to a greater degree), the expression of PKA-C and most other isoforms was not affected (*Figure 1—figure supplement 1*), the localization of the modified PKAR-IIβ was confined to the cytosol prior to stimulation (*Figure 1C*) and there was no detectable phenotypic effect of this induced expression.

To assay translocation of PKA-R to the membrane, HeLa cells transiently expressing the mCherry-tagged translocation system were treated with 100 nM rapamycin and imaged for 30 min. Addition of rapamycin resulted in rapid PKA-R translocation to the PM evaluated as a significant decrease in the cytoplasmic fluorescence intensity over the first 3 min of stimulation (*Figure 1C and D*). No significant translocation occurred after this initial period. To determine whether PKA-C translocated to the membrane along with the exogenous PKA-R, we co-transfected HeLa cells with PKAR-FKBP-YFP, Lyn-FRB, and the constructs encoding either recombinant mCherry protein or mCherry-tagged PKA-C. Following addition of rapamycin, PKA-C indeed translocated to the cell membrane whereas mCherry alone underwent no significant change in localization, demonstrating that our synthetic system can bring the intact PKA holoenzyme to the PM (*Figure 1E* and *Videos 1 and 2*). Furthermore, in contrast to absence of any detectable phenotypic effect of PKAR-FKBP-FP expression in unstimulated HeLa cells, these cells treated with rapamycin displayed a rapid change in cell morphology (spreading) and an increase in filopodia formation (not observed in untransfected cells treated with the same rapamycin dose) (*Figure 1C* and *Video 1*), pointing to a pronounced effect of a change in the local PKA-R levels at the PM.

### Characterization of cell response to PKA-R translocation

The dissociation of PKA-C from PKA-R is commonly seen as a relief of PKA-C inhibition within the holoenzyme, with the regulatory subunit thus treated as a negative regulator of the kinase activity. To test whether membrane translocation of PKA-R would indeed inhibit the basal PKA activity in this compartment, we transfected HeLa PFM cells with Lyn-AKAR4, a membrane-bound FRET probe for PKA activity (*Depry et al., 2011*). The dynamic range of the intracellular Lyn-AKAR4 responses in this cell line was determined to be 20.9% ± 2.0% ($n$ = 12) (mean ± standard error of the mean [SEM] [$n$ = number of cells]) following cell treatment with a combination of an adenylyl cyclase activator forskolin (Fsk, 50 µM) and a competitive non-selective phosphodiesterase inhibitor 3-isobutyl-1-methylxanthine (IBMX, 100 µM) to maximally increase the intracellular cAMP levels (*Figure 2—figure supplement 1*). Surprisingly, we found that rapamycin-induced PKA-R translation alone (without any additional stimulation) was able to induce a significant increase in PKA activity, which was transient in some cells and sustained in others (*Figure 2A and B*). To determine whether the variability in the concentration of PKA-R could account for this cell-cell variation of response, we examined the dependence of the maximum levels of PKA activity on the estimated PKA-R abundance, using mCherry fluorescence intensity as a proxy. Interestingly, we observed that the increase in activity was greatest for cells with

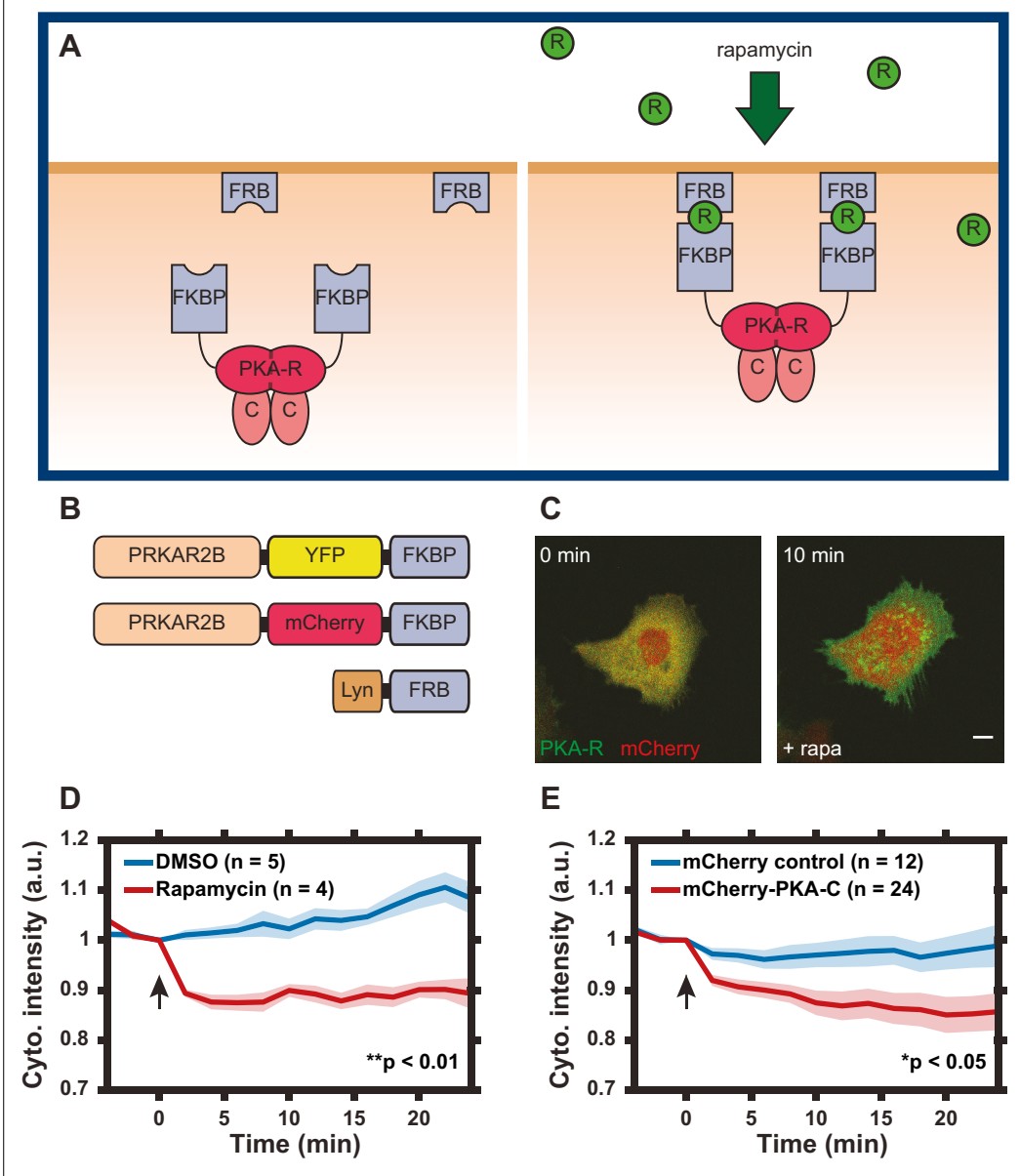

**Figure 1.** Design of regulatory subunit of the protein kinase A (PKA-R) translocation system. (**A**) Schematic of PKA-R translocation system. Rapamycin induces heterodimerization of FKBP and FRB, resulting in translocation of PKA-R to the plasma membrane (FKBP = FK506-binding protein, FRB = FKBP-rapamycin-binding domain, R = rapamycin, PKA-R = PKA regulatory subunit, C = PKA catalytic subunit). (**B**) DNA construct design. Two versions of recombinant PKA-R were created with different fluorescent labels. (**C**) Subcellular localization of PKAR-FKBP-YFP (green) within a transiently transfected HeLa cell at 0 and 10 min after addition of 100 nM rapamycin. Scale bar, 10 µm. mCherry protein (red) co-expressed for visualization. (**D**) PKA-R translocation in HeLa PFM cells quantified as cytoplasmic intensity drop in mCherry channel following addition of DMSO or 100 nM rapamycin. p = 0.0039 at $t$ = 24 min post-rapamycin addition; two-tailed Student's $t$-test. (**E**) Catalytic subunit of the protein kinase A (PKA-C) translocation in HeLa cells transiently transfected with PKAR-FKBP-YFP, Lyn-FRB, and mCherry-PKA-C, quantified as a cytoplasmic intensity drop in mCherry channel following addition of 100 nM rapamycin. Cells transfected with mCherry protein instead of mCherry-PKA-C were used as a control. p = 0.037 at $t$ = 24 min post-rapamycin addition; two-tailed Student's $t$-test. Graphs display the mean of each data set with standard error of the mean (SEM) indicated by shaded region. Number of cells in each data set is as indicated in the figure. Data is the result of one (**D**) and three (**E**) independent experiments, respectively. Mean and SEM values for each condition and time point are provided in *Figure 1—source data 1*. Arrows indicate the timing of drug addition.

The online version of this article includes the following source data and figure supplement(s) for figure 1:

*Figure 1 continued on next page*

*Figure 1 continued*

**Source data 1.** Characterization of regulatory subunit of the protein kinase A (PKA-R) translocation system.

**Figure supplement 1.** Immunoblotting for regulatory subunit of the protein kinase A (PKA-R) isoforms and catalytic subunit of the protein kinase A (PKA-C).

**Figure supplement 1—source data 1.** Raw immunoblot images, labeled and unlabeled.

**Figure supplement 1—source data 2.** Immunoblot statistical analysis.

**Figure supplement 2.** Plasmid maps for PKAR-FKBP-FP constructs.

intermediate PKA-R concentrations, decreasing when PKA-R was either higher or lower than this optimal level (*Figure 2C*). Furthermore, in the cells with the highest PKA-R expression, PKA activity, following the initial rise, later not only decreased vs. the maximum, but dropped below the basal level, indicating active inhibition of the PKA activity (*Figure 2D and E*). Notably, even in the cells in which the PKA activity was inhibited vs. the basal levels, this activity transiently increased, suggesting that the PKA-R translocation was gradual, and thus first reached activating levels at the PM, but ultimately exceeded these levels and reached inhibitory concentrations.

To further investigate whether the local concentration of PKA-R played a role in determining the magnitude and duration of the response at the PM, we treated cells with two different lower concentrations of rapamycin – 2 and 20 nM. We again found that treatment of cells with 20 nM of rapamycin led to PKA activation that was, on average, transient, consistent with the gradual accumulation of PKA-R at the PM, first to optimal and then inhibitory levels. Importantly, we found that inducing a decreased level of PKA-R translocation with a lower dose of rapamycin (2 nM) resulted in a slower but much more sustained increase in PKA activity, reaching, on average, much higher levels than those seen for the higher dose (*Figure 2F*), suggesting that the lower PKA-R levels achieved at this rapamycin concentration were close to optimal. These results collectively suggested that PKA-R translocation to the PM can stimulate the PKA activity up to an optimal level of this subunit but can also inhibit PKA when the PKA-R levels exceed the optimal level in the PM compartment.

To account for this paradoxical effect of the local PKA-R abundance, we hypothesized that similar to linker or scaffold molecules, this subunit can both enhance and inhibit signaling output, depending on its concentration relative to the concentrations of the other components of the holoenzyme (e.g., cAMP or PKA-C molecules). Over-abundance of this molecule can 'titrate out' other complex

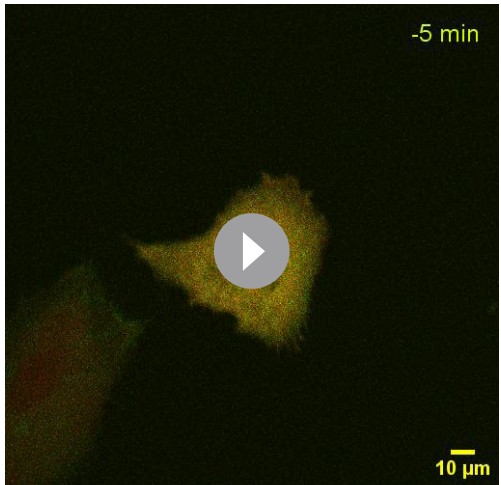

**Video 1.** Rapamycin-induced translocation of regulatory subunit of the protein kinase A (PKA-R). Translocation of PKA-R (green) to the plasma membrane of HeLa cells following treatment with 100 nM rapamycin at time zero as indicated in the video. Cells were co-transfected with mCherry protein (red) as a counterstain for visualization.

https://elifesciences.org/articles/66869/figures#video1

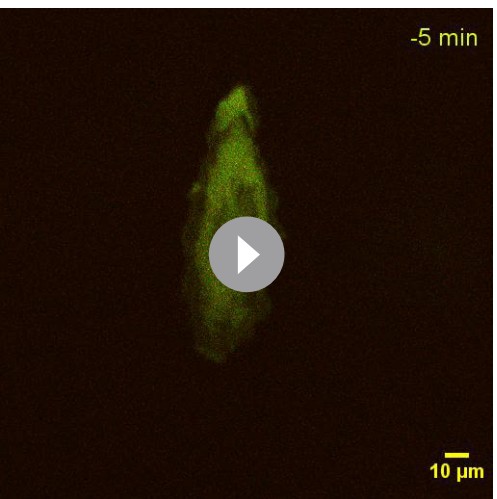

**Video 2.** Co-recruitment of catalytic subunit of the protein kinase A (PKA-C) and regulatory subunit of the protein kinase A (PKA-R) to the *plasma membrane*. Co-localization of PKA-C (red) and PKA-R (green) before and after treatment with 100 nM rapamycin at time zero as indicated in the video.

https://elifesciences.org/articles/66869/figures#video2

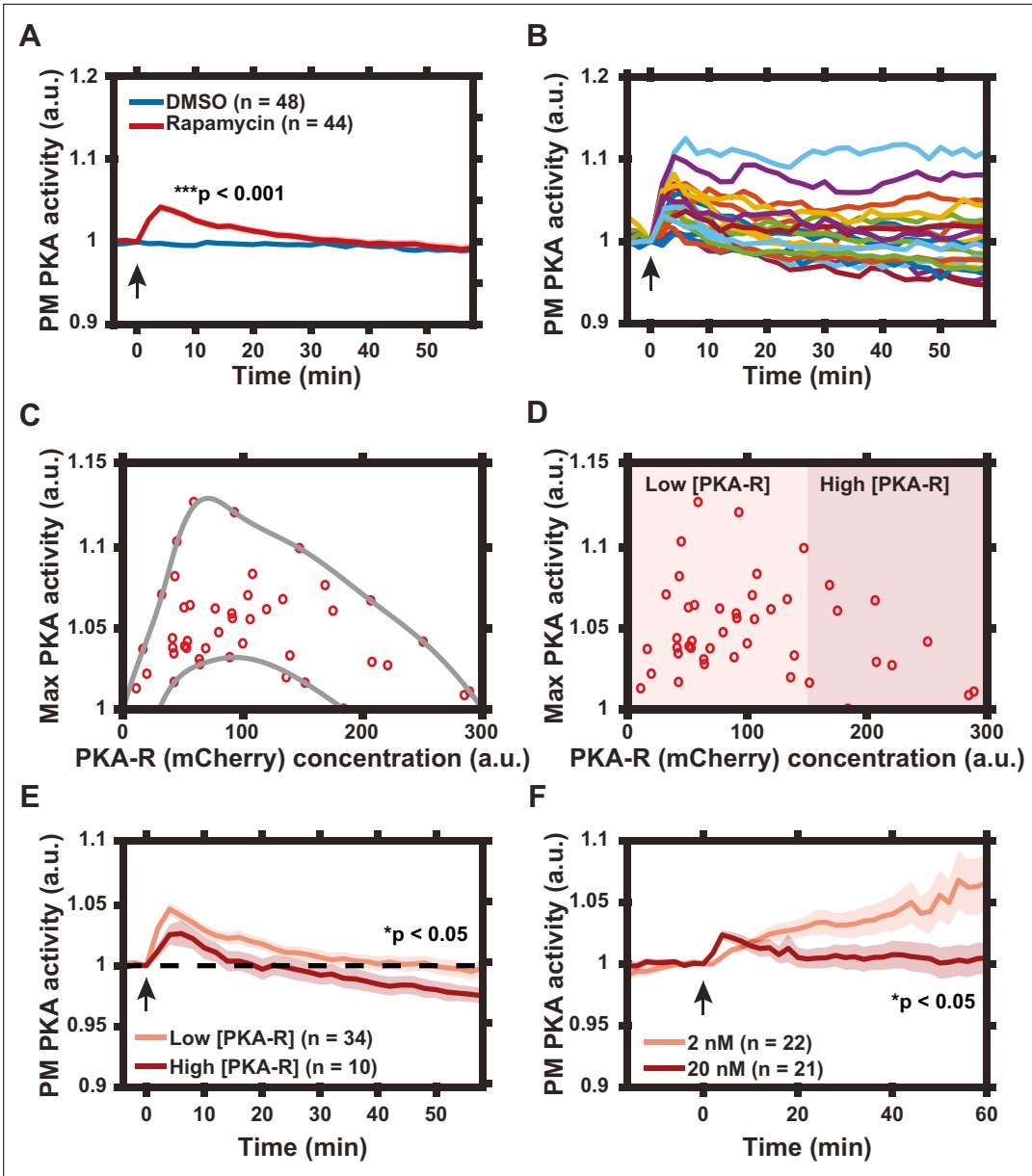

**Figure 2.** Characterization of protein kinase A (PKA) activity response to regulatory subunit of the PKA (PKA-R) translocation. (**A**) Effect of rapamycin-induced PKA-R translocation on PKA activity at the plasma membrane (PM) as detected by Lyn-AKAR4. 100 nM rapamycin or 0.1% DMSO added at time = 0. p = 9.83 x $10^{-12}$ at $t$ = 4 min post-rapamycin addition; two-tailed Student's $t$-test. (**B**) Single cell PKA activity dynamics at the PM following addition of 100 nM rapamycin (a subset of data used for (**A**) is shown for clarity). (**C**) Relationship between PKA-R concentration, as estimated by mCherry fluorescence intensity, and maximal PM PKA activity increase following PKA-R translocation (n = 44 cells). Envelope overlaid for visualization. (**D**) Classification of cells into 'low' and 'high' expressors of PKA-R. (**E**) Average PM PKA activity over time for 'low' vs. 'high' expressors of PKA-R (defined in panel D). p = 0.024 at $t$ = 58 min post-rapamycin addition; two-tailed Student's $t$-test. (**F**) PM PKA activity response to two different rapamycin doses. p = 0.023 at $t$ = 60 min post-rapamycin addition; two-tailed Student's $t$-test. Graphs in (**A, E, F**) display the mean of each data set with standard error of the mean (SEM) indicated by shaded region. Number of cells in each data set is as indicated in the figure. Data is the result of three (**A–E**) and two (**F**) independent experiments, respectively. Mean and SEM values for each condition and time point, as well as single cell measurements for data presented in (**A–E**), are provided in *Figure 2—source data 1*. Arrows indicate the timing of drug addition. All experiments completed in HeLa PFM cells transiently expressing Lyn-AKAR4.

The online version of this article includes the following source data and figure supplement(s) for figure 2:

*Figure 2 continued on next page*

*Figure 2 continued*

**Source data 1.** Lyn-AKAR4 data following regulatory subunit of the protein kinase A (PKA-R) translocation.

**Figure supplement 1.** Lyn-AKAR4 dynamic range.

components thus exerting negative effect, whereas the lower levels of the linker are essential for the complex functionality. We constructed a new heuristic mathematical model of this effect, which, along with our prior modeling effort (*Levchenko et al., 2000*), suggested that there is indeed an optimal level of concentration expected for PKA-R, if it behaves as such a linker molecule (Appendix 1). Furthermore, an increase in the local PKA-R abundance was predicted to potentially drive the signaling to below baseline levels, if the initial concentration of PKA-R at the PM was sufficiently high prior to forced PKA-R translocation.

Overall, the combination of experimental and mathematical analyses suggested that rapamycin-based PKA-R translocation to the PM leads to a gradual accumulation of this PKA subunit in the specific subcellular compartment, allowing it to progressively accumulate at sub-optimal, optimal, and supra-optimal levels, and thus initially enhance and then inhibit signaling output, by virtue of serving the role of a linker in the holoenzyme complex.

## Graded translocation of PKA-R induces a reversal of cell polarity

We next explored the functional effects of induced PM localization of PKA-R. PKA activity and PKA-R abundance have been shown to be elevated at the front of migrating cells in vitro, but the role that spatial localization of PKA plays in regulating cell migration is not well understood (*Lim et al., 2008*; *Paulucci-Holthauzen et al., 2009*). We therefore used our translocation system to probe the effect of intracellular PKA activity gradients, induced by gradients of PKA-R PM translocation, on cell polarization and migration. To accomplish this, we took advantage of the fact that intracellular PKA-R-FKBP-FP gradients can be induced by extracellular rapamycin gradients controlled within a microfluidic device (*Lin et al., 2012*; *Lin and Levchenko, 2015*; *Lin and Levchenko, 2015*). We used a variant of the microfluidic chip capable of generating chemical gradients by passive diffusion, which was previously developed in our lab (*Lin et al., 2015*). We increased the throughput to enable observation of up to 304 individual cells in parallel narrow microchannels ('cross-channels') allowing for 1D cell migration (*Figure 3A* and Appendix 2). A gradient of rapamycin in the cross-channels was induced through diffusion between continuously replenished 'source' and 'sink' channels and visualized using a dye of similar molecular weight (Alexa Fluor 594). The cells in the channels exposed to the gradient did not experience sheer stress because of a much higher hydraulic resistance within the channels relative to a much lower resistance in larger source and sink channels. HeLa PFY cells were seeded into the chip and allowed to migrate into cross-channels overnight. Due to directionality of cell seeding, most cells had an initial directional polarity of migration, resulting in the 'upward' migration direction (from 'sink' to 'source', as labeled in *Figure 3A*) before rapamycin was added.

Upon application of a rapamycin gradient (0–20 nM, 0.08 nM/µm), we found a pronounced reversal of the directionality of the cells' migration in the direction opposite to the initial 'upward' bias, with the preferred new direction thus being opposite to the direction of the rapamycin gradient (*Figure 3C and E* and *Video 3*). This was in stark contrast to cells without expression of Lyn-FRB or cells exposed to a DMSO gradient that did not experience PKA-R translocation and continued migrating in the 'upward' direction (*Figure 3C and D*, *Figure 3—figure supplement 1*, and *Video 4*). This result was surprising and in apparent contradiction with the expectation that an increase in PKA-R at the cell front would enhance the pre-existing cell polarization toward the rapamycin source. However, it was consistent with our prior observations suggesting that in many cells, the translocation of PKA-R induced by a sufficiently high rapamycin concentration could have an inhibitory rather than activating effect on PKA. This inhibitory effect was certainly true for a 20 nM spatially homogeneous dose of rapamycin, as revealed by the experiments and analysis described above (*Figure 2F*).

To determine whether PKA activity was indeed affected by a graded translocation of PKA-R, we repeated the experiment with HeLa PFM cells transiently expressing Lyn-AKAR4. Prior to rapamycin exposure, most cells displayed an internal PKA activity gradient that corresponded with the cell polarization state (high PKA activity at the cell front and low at the rear) (*Figure 3F*). Upon exposure to the rapamycin gradient, we observed a reversal in the direction of the internal PKA activity gradient that was concurrent with the flip in the direction of cell migration (*Figure 3F and G*). Interestingly, we also

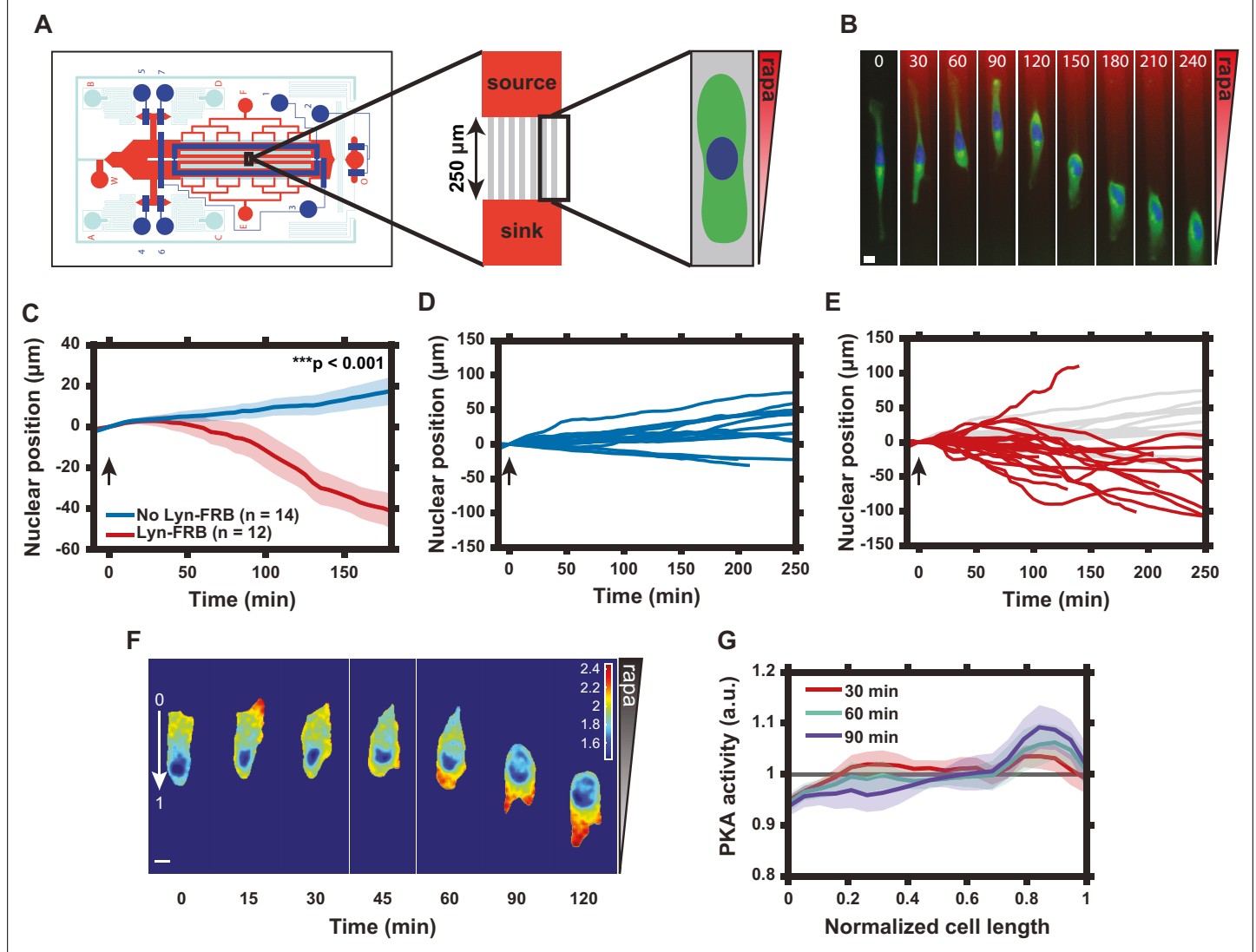

**Figure 3.** Graded translocation of regulatory subunit of the protein kinase A (PKA-R) induces a reversal of cell polarity. (**A**) Schematic of the microfluidic device used to produce gradients of rapamycin across microchannels housing HeLa PFY or PFM cells. (**B**) Single cell response to 20 nM rapamycin gradient. Numbers show time in minutes. (Green = PKAR-FKBP-YFP [stably expressed], Blue = H2β-mCerulean [transiently expressed nuclear marker], Red = Alexa Fluor 594 dye). (**C**) Average nuclear position (normalized to t = 0) for HeLa PFY cells in 20 nM rapamycin gradient (0.08 nM/μm) vs. no-translocation rapamycin control (HeLa cells stably expressing PKAR-FKBP-YFP but not Lyn-FRB). Standard error of the mean (SEM) indicated by shaded regions. p = 1.34 × 10⁻⁵ at 180 min post-rapamycin addition; two-tailed Student's t-test. Number of cells in each data set as indicated in the figure. (**D**) Single cell nuclear position data for no-translocation rapamycin control cells in 20 nM rapamycin gradient. Data from one independent experiment. (**E**) Single cell nuclear position data for HeLa PFY cells in 20 nM rapamycin gradient. Data from three independent experiments. Data from (**D**) superimposed in light gray. (**F**) Tracking of intracellular PKA activity gradient in HeLa PFM cells using the transiently expressed FRET probe Lyn-AKAR4. Rapamycin gradient introduced at t = 0. Colorimetric FRET ratio scale as indicated by color bar. (**G**) Mean intracellular plasma membrane (PM) PKA activity tracking along the cell length from the high end of the rapamycin gradient ('0' in panel F) to the low end ('1' in panel F). Data represent the mean of n = 9 HeLa PFM cells from two independent experiments with SEM indicated by shaded region. Cells were divided into 20 bins with the average FRET ratio value taken for each. Arrows in (**C–E**) indicate addition of rapamycin. Scale bars in (**B**) and (**F**), 10 μm. Mean and SEM values for each time point and condition in (**C**) and each time point and position in (**G**) are provided in *Figure 3—source data 1*.

The online version of this article includes the following source data and figure supplement(s) for figure 3:

**Source data 1.** Nuclear position and Lyn-AKAR4 data in microfluidic device.

**Figure supplement 1.** 1D cell migration in DMSO gradient.

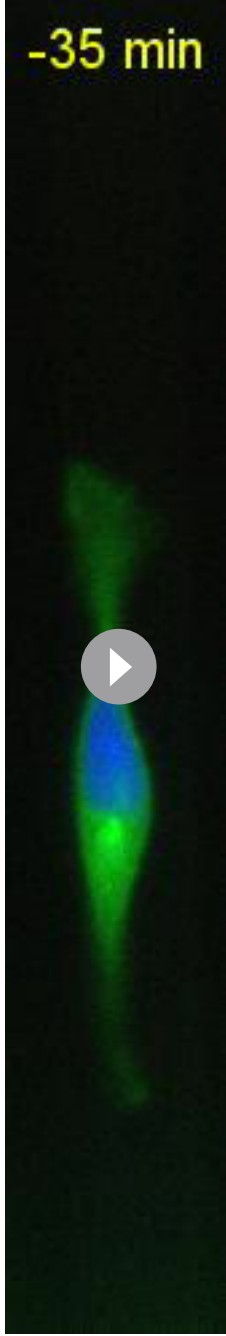

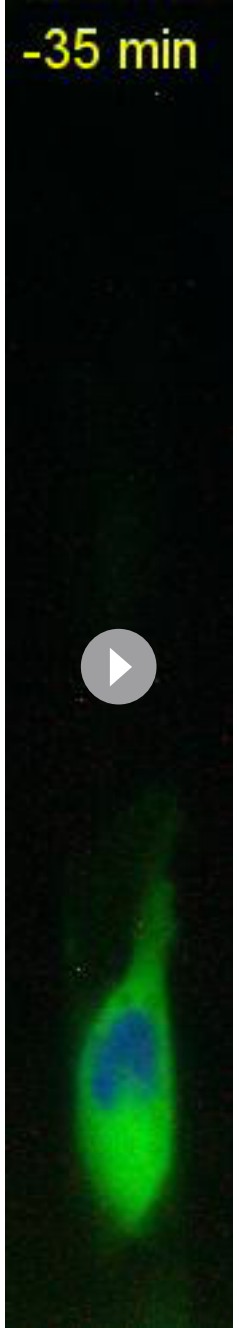

**Video 3.** Single cell response to graded regulatory subunit of the protein kinase A (PKA-R) recruitment. Single cell response to 20 nM rapamycin gradient applied at time zero as indicated in the video. (Green = PKAR-FKBP-YFP [stably expressed], Blue = H2β-mCerulean [transiently expressed nuclear marker], Red = Alexa Fluor 594 dye).

https://elifesciences.org/articles/66869/figures#video3

**Video 4.** Single cell response to DMSO gradient. Single cell response to DMSO gradient applied at time zero as indicated in the video. (Green = PKAR-FKBP-YFP [stably expressed], Blue = H2β-mCerulean [transiently expressed nuclear marker], Red = Alexa Fluor 594 dye).

https://elifesciences.org/articles/66869/figures#video4

observed a transient increase in PKA activity at the cell front (*Figure 3F*, 15 min time point) prior to this reversal. These results, in combination, further supported the explanation of the reversal of cell migration directionality due to a transiently positive but, in the longer run, negative effect of a high level of PKA-R PM translocation.

## Cell polarization state can be tuned by the slope of the rapamycin gradient

To further explore the mechanism of the reversal of cell migration directionality, we investigated how the slope and magnitude of the rapamycin gradient affected the response of HeLa PFY cells. First, we explored the effects of spatially uniform (20 nM – 20 nM and 100 nM – 100 nM across the channel) rapamycin inputs. Both were expected to compete with internal polarity cues. We found that whereas the lower rapamycin concentration had no detectable effect (vs. e.g., the control *Figure 3D*), the higher concentration gradually randomized the cell migration directionality, suggesting a stronger negating effect on the intrinsic polarity regulation (*Figure 4A*). We then exposed the cells to a relatively shallow rapamycin gradient (10 nM – 20 nM across the channel, or 0.04 nM/μm), contrasting the results with those produced by steeper spatially graded stimulation (0–20 nM, or 0.08 nM/μm, *Figure 3E*). We again found reversal of the average migration directionality, which was increasingly more pronounced with increasing gradient slope (*Figure 4D and F*). This result suggested that both gradient values were sufficient to compete with the intrinsic polarity regulation mechanisms, by suppressing PKA activity in the front of the cell more than in the rear. By this logic, we expected that presenting cells with a reverse gradient (20–0 nM, 'downward') would result in internal gradients of PKA activity that would effectively point 'upward' for a large subset of cells, which would be consistent with the directionality of their inherent polarity, an effect opposite to the rapamycin gradients pointing 'upward'. Thus, cells were expected to continue moving 'upward', perhaps at an even greater rate vs. the control. We indeed observed that most cells under this condition moved similarly to the control case, while a subset of cells reversed their directionality from 'downward' to 'upward', and another subset displayed an increased speed of 'upward' migration vs. the maximal cell migration levels observed in the control (*Figure 4G and H*).

## Discussion

Proper kinase localization is important for cell function, ensuring that the kinase activity is limited to a specific set of substrates in response to an extracellular cue. Here, we describe development and implementation of a novel approach to dynamically relocalize a regulatory subunit of a kinase, PKA, to the PM. In contrast to a previous relocalization technique utilizing a photoactivatable PKA-C, our approach maintains the dependence of resulting PKA activity on cAMP and does not require overexpression of the catalytic subunit (*O'Banion et al., 2018*). Furthermore, our synthetic system recapitulates the natural mechanism of subcellular PKA targeting of the regulatory PKA subunits by AKAPs.

In regulation of PKA activity, the regulatory PKA subunit can play a dual role of an inhibitor of PKA-C and a mediator of PKA localization to specific subcellular compartments, potentially enriched in the kinase activator (cAMP) and kinase substrates. Thus, in spite of its canonically inhibitory function, PKA-R can potentially have a more positive, up-regulating role on the activity of the enzyme. Our approach has allowed us to clarify this paradoxical function of PKA-R. We demonstrate in particular that relocalization of PKA-R to the PM was sufficient to induce an increase in PKA activity in this intracellular compartment, particularly for lower levels of the translocated PKA-R, even in the absence of changes in upstream signaling through G protein-coupled receptors. The effect of the PKA-R translocation reached the maximal level at the optimum level of this subunit, decreasing at PKA-R higher levels and ultimately driving the PKA activity to below basal levels. This type of nonlinear behavior has been experimentally and computationally demonstrated for linker proteins, such as scaffold proteins (*Chapman and Asthagiri, 2009*; *Good et al., 2011*; *Levchenko et al., 2000*). At low levels, a scaffold can enhance signaling by bringing components of a signaling pathway into close proximity, enabling their interaction, and thus enhancing the pathway activity. However, when the scaffold concentration is too high, it sequesters pathway components away from each other, decreasing the pathway activity, a behavior that is recapitulated by our mathematical model. Our results suggest that within the cell, PKA-R plays a scaffold-like role for PKA-C and cAMP since the kinase activity requires linkage of two PKA-C subunits and four cAMP molecules into one molecular complex by these subunits. When the local PKA-R concentration is too high, it can lead to incomplete binding (titration away) of either cAMP or PKA-C, preventing activation. Therefore, as PKA-R is recruited to the membrane, PKA activity will increase until there is a local shortage of one or both of these components. This suggests a more complex view of PKA-R's role in this pathway, suggesting that it can have either a pathway inhibitory

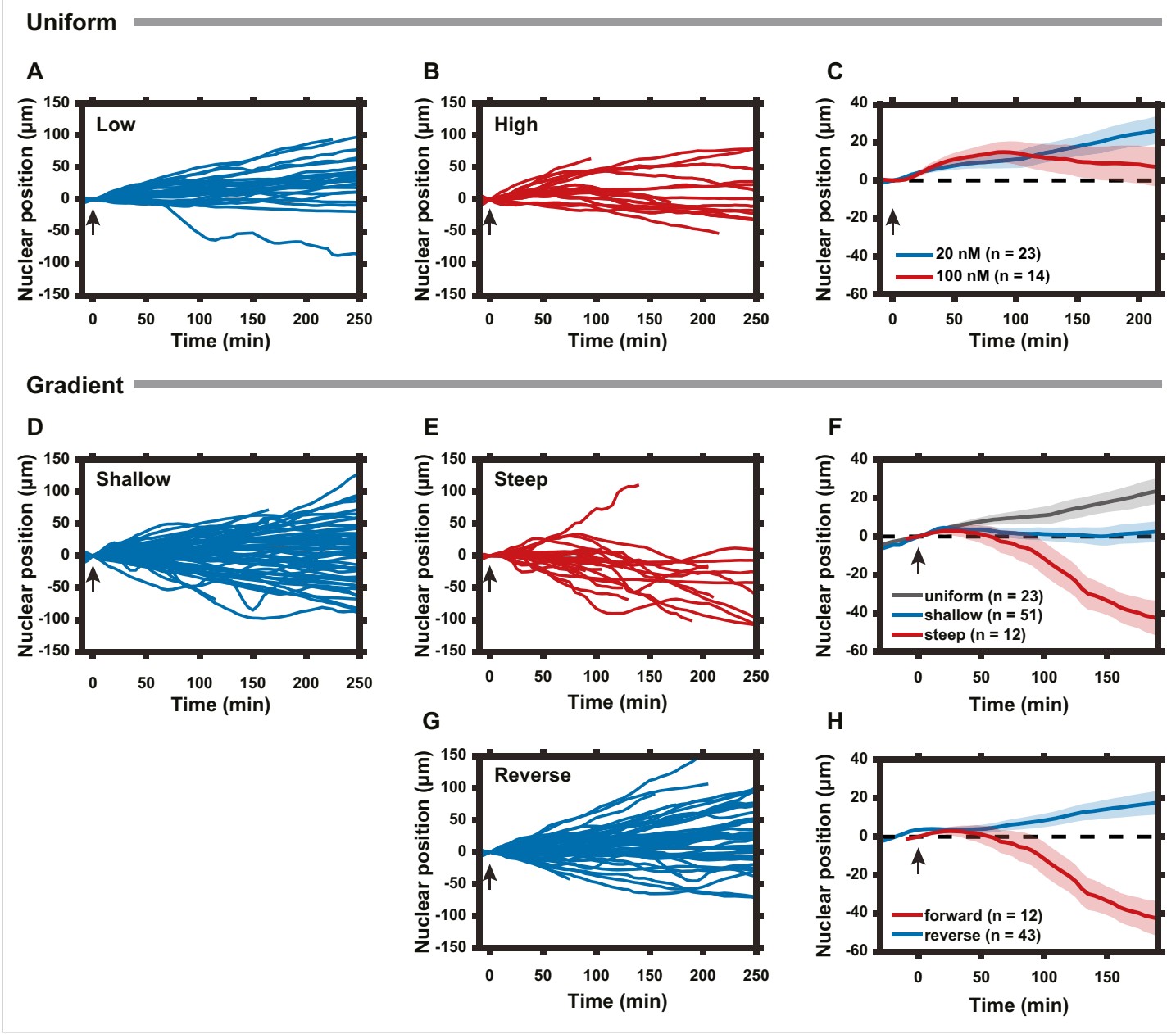

**Figure 4.** Cell polarization state can be tuned by the slope of the regulatory subunit of the protein kinase A (PKA-R) gradient. (**A**) Nuclear positions of single cells responding to a uniform 20 nM rapamycin stimulus. (**B**) Nuclear positions of single cells responding to a uniform 100 nM rapamycin stimulus. (**C**) Average nuclear position of cells responding to 20 nM vs. 100 nM uniform rapamycin stimulus. Standard error of the mean (SEM) indicated by shaded regions. (**D**) Nuclear positions of single cells responding to a shallow rapamycin gradient (10 nM -– 20 nM, or 0.04 nM/μm). (**E**) Nuclear positions of single cells responding to a steep rapamycin gradient (0 nM -– 20 nM, or 0.08 nM/μm). Data taken from *Figure 3E*. (**F**) Average nuclear position of cells responding to a uniform 20 nM stimulus from (**A**), shallow gradient from (**D**), or steep gradient from (**E**). Standard error of the mean (SEM) indicated by shaded regions. (**G**) Nuclear positions of single cells responding to a steep rapamycin gradient in the reverse direction (20 nM – 0 nM, or 0.08 nM/μm) as what is shown in (3A). (**H**) Average nuclear position of cells responding to steep rapamycin gradient in the forward and reverse directions. SEM indicated by shaded regions. Arrows indicate addition of rapamycin. Number of cells in (**C**), (**F**), and (**H**) are as indicated in the figure. Data was collected from two (**A, D, G**) and three (**B**) independent experiments. Mean and SEM values for each time point and condition are provided in *Figure 4—source data 1*. All experiments completed using HeLa PFY cells.

The online version of this article includes the following source data for figure 4:

**Source data 1.** Nuclear position data for cells exposed to different rapamycin stimuli in microfluidic device.

or enhancing role, depending on the expression of PKA-C or cAMP abundance both globally across the cell and in specific subcellular compartments. In particular, our results suggest that the abundance of PKA-R at the PM in the HeLa cells investigated here is sub-optimal, but one can expect that it can be optimal or 'super-optimal' in other cellular compartments.

Our tool also allowed us to study the functional effect of controlling the basal PKA activity on a complex phenotypic response: the polarity of cell migration. Although PKA-R has been shown to be enriched at the leading edge (*Howe et al., 2005*) of moving cells, the functional importance of either this localization pattern or the overall PKA activity gradient across a polarized cell has not been fully understood. Indeed, PKA activity gradients can be triggered by a number of membrane localized receptors or other signaling proteins, which are frequently pleiotropic and can activate other signaling pathways. Within this in vivo complexity, our molecular tool can isolate the specific role of PKA activity in defining cell polarity. It can be particularly revealing if one can reverse the cell polarity by directly imposing an intracellular PKA activity gradient. We indeed found that inducing gradients of PKA-R PM translocation at levels inhibitory to PKA activity led to a reversal of the internal PKA activity gradient and, as a result, the cell polarity and direction of cell migration. This work therefore demonstrates for the first time that intracellular PKA activity gradients can provide a polarization cue that is powerful enough to overcome other existing polarization cues inside the cell.

It is likely that PKA activity gradients exert an effect on cell polarization via modulation of the activity of the two mutually inhibitory Rho-family small GTPases RhoA and Rac1. PKA phosphorylation of RhoA on Ser188 has been shown to promote its interaction with RhoGDI, leading to removal of RhoA from the PM (*Lang et al., 1996*). PKA is also implicated in Rac1 activation, although the direct target of this positive regulation is still unclear (*O'Connor and Mercurio, 2001*). Since PKA has a positive effect on Rac1 activity and a negative effect on membrane localized RhoA activity, it would be reasonable to theorize that PKA acts to shift the balance of Rac1 to RhoA signaling in favor of Rac1. In this way, a gradient of intracellular PKA activity could help to enhance the internal polarization of these two polarity effectors. When the PKA activity gradient flips, as in *Figure 3G*, the effect would be to increase Rac1 and decrease RhoA activity at the cell rear, leading to a reversal of polarity if this signal is sufficiently strong.

In a broader sense, this work demonstrates the critical role that kinase localization plays in controlling the cell response to an incoming stimulus. The subcellular kinase concentration can be tuned by the expression levels of various scaffold proteins, and differential expression between cells may result in variability of signaling responses within a population. Localization of scaffold proteins may vary dynamically via modifications to subcellular localization sequences as well, resulting in similar effects to those described here. In the context of PKA, AKAPs are receiving attention as potential therapeutic targets, and at least one isoform of PKA-R has been implicated as a driver of oncogenesis (*Codina et al., 2019*; *Esseltine and Scott, 2013*; *Veugelers et al., 2004*). The kinase relocalization strategy demonstrated herein can be applied to study the effects of localized PKA activity in other subcellular compartments and can be adapted to the study of other kinases, enabling a better understanding of the role of compartmentalized signaling.

## Materials and methods

**Key resources table**

| Reagent type (species) or resource | Designation | Source or reference | Identifiers | Additional information |
|---|---|---|---|---|
| Gene (*Homo sapiens*) | PRKAR2B | GenBank | NCBI Reference Sequence:NM_002736.2 | mRNA sequence for PKA RIIβ |
| Strain, strain background (*Escherichia coli*) | DH5α | ThermoFisher Scientific | Cat#:18263012 | Competent cells |
| Cell line (human) | HeLa | Gift from Dr Jin Zhang, UCSD | RRID:CVCL_0030 | |
| Cell line (human) | HeLa PFY | This study | | See 'Cell line generation' in Materials and methods |
| Cell line (human) | HeLa PFM | This study | | See 'Cell line generation' in Materials and methods |

*Continued on next page*

*Continued*

| Reagent type (species) or resource | Designation | Source or reference | Identifiers | Additional information |
|---|---|---|---|---|
| Cell line (human) | HEK293FT | Laboratory stock | RRID:CVCL_6911 | Used to generate lentivirus for creation of cell lines |
| Antibody | Anti-PKA[C] (Mouse monoclonal) | BD | Cat#:610981; RRID: AB_398293 | (1:1000, WB) |
| Antibody | Anti-PKA RIIβ (Mouse monoclonal) | Santa Cruz Biotechnology | Cat#:sc-376778 | (1:400, WB) |
| Antibody | Anti-PKA RIIα (Rabbit polyclonal) | Abcam | Cat#:ab38949 | (1:1000, WB) |
| Antibody | Anti-PKA RI-α/β (Rabbit polyclonal) | Cell Signaling Technology | Cat#:3927 | (1:1000, WB) |
| Antibody | Anti-GAPDH (Rabbit monoclonal) | Cell Signaling Technology | Cat#:2118 | (1:1000, WB) |
| Recombinant DNA reagent | PKAR-FKBP-YFP transient expression plasmid | This study | | See 'Plasmid design' in Materials and methods |
| Recombinant DNA reagent | PKAR-FKBP-mCh transient expression plasmid | This study | | See 'Plasmid design' in Materials and methods |
| Recombinant DNA reagent | Lyn₁₁-FRB transient expression plasmid | DOI:10.1126/ science.1131163 | | Gift from Dr Takanari Inoue, Johns Hopkins University |
| Recombinant DNA reagent | pLenti-cmv-PKAR-FKBP-YFP-puro dest | This study | | Lentiviral construct for expression of PKAR-FKBP-YFP. See 'Cell line generation' in Materials and methods |
| Recombinant DNA reagent | pLenti-cmv-PKAR-FKBP-mCh-puro dest | This study | | Lentiviral construct for expression of PKAR-FKBP-mCh. See 'Cell line generation' in Materials and methods |
| Recombinant DNA reagent | pLenti-cmv-Lyn-FRB-blast dest | This study | | Lentiviral construct for expression of Lyn₁₁-FRB. See 'Cell line generation' in Materials and methods |
| Recombinant DNA reagent | Lyn-AKAR4 | DOI:10.1039/ c0mb00079e | | Gift from Dr Jin Zhang, UCSD |
| Recombinant DNA reagent | H2β-mCerulean | This study | | See 'Plasmid design' in Materials and methods |
| Commercial assay or kit | QIAprep Spin Miniprep kit | QIAGEN | Cat#:27104 | |
| Commercial assay or kit | Pierce BCA Protein Assay Kit | ThermoFisher Scientific | Cat#:23225 | |
| Chemical compound, drug | Rapamycin | LC Laboratories | Cat#:R-5000 | |
| Chemical compound, drug | Forskolin; Fsk | MilliporeSigma | Cat#:344270; CAS:66575-29-9 | |
| Chemical compound, drug | 3-Isobutyl-1-methylxanthine; IBMX | MilliporeSigma | Cat#:I5879; CAS:28822-58-4 | |
| Chemical compound, drug | Fugene HD Transfection Reagent | Promega | Cat#:E2311 | |
| Chemical compound, drug | TurboFect Transfection Reagent | ThermoFisher Scientific | Cat#:FERR0531 | |
| Software, algorithm | MATLAB, R2021b and prior versions | MathWorks | | Used for custom image analysis. Code provided with manuscript |
| Software, algorithm | GraphPad Prism 9.3.1 | GraphPad Software, LLC | | Used for statistical analysis |
| Software, algorithm | Fiji/ImageJ | DOI: 10.1038/ nmeth.2019 | RRID:SCR_002285 | Used for immunoblot analysis |
| Other | Multilayer microfluidic device/ gradient generator | This study | | See Materials and methods and Appendix 2 |

## Cell culture and transfection

HeLa cells were cultured in high glucose Dulbecco's modified Eagle medium (DMEM, Corning, Corning, NY) supplemented with 10% fetal bovine serum (FBS) (ThermoFisher Scientific, Waltham, MA) and 1% penicillin/streptomycin (ThermoFisher Scientific). For stable lines, media was supplemented with 1 µg/ml puromycin (MilliporeSigma, Burlington, MA) and blasticidin (ThermoFisher Scientific) to

continuously select for cells expressing PKAR-FKBP-FP and Lyn₁₁-FRB (also referred throughout to as Lyn-FRB). Cells were cultured in a 37°C humidified incubator with 5% $CO_2$ and kept isolated from other cell lines while in culture to prevent cross-contamination. DAPI staining was performed to test for mycoplasma contamination. The HeLa cell line was not authenticated over the course of this study since the significance of the results is independent of the specific cell source. Relevant characteristics that are cell line-dependent were specifically tested (for instance, PKA regulatory subunit expression levels). Transient transfections were completed using Fugene (Promega, Madison, WI) according to manufacturer's protocol. Transfection of lentiviral vectors was completed using TurboFect (ThermoFisher Scientific).

## Plasmid design

RNA was isolated from HeLa cell lysate, reverse transcribed, and PCR amplified for the cDNA sequences of PKA-RIIβ and PKA-Cβ respectively using custom primers. The gel purified PCR product was ligated into a transient expression plasmid containing FKBP and either YFP or mCherry fluorescent protein (gifts from Dr Takanari Inoue, Johns Hopkins University) in the case of PKAR-FKBP-FP, or mCherry alone in the case of mCherry-PKA-C. The plasmid was then transformed into DH5α competent cells, amplified, recovered using a standard QIAprep Spin Miniprep kit (Qiagen, Hilden, Germany), and sent for sequencing. The H2β-mCerulean transient expression plasmid was created by cloning the H2β DNA sequence into the mCerulean-N1 vector (Addgene #54758). The plasmid was transformed and amplified in DH5α cells, recovered as above, and sent for sequencing.

## Cell line generation

PKAR-FKBP-FP and Lyn-FRB were transferred to lentiviral expression vectors by Gateway cloning. Both color variants of PKAR-FKBP-FP were cloned into pLenti CMV puro destination vectors (Addgene plasmid #17452). Lyn-FRB was cloned into the pLenti CMV blast destination vector (Addgene plasmid #17451). To generate lentivirus, HEK293FT cells were transfected with one of the three destination vectors plus a lentiviral packaging vector (psPAX2, Addgene plasmid #12260) and a VSV-G envelope expressing vector (pMD2.G, Addgene plasmid #12259). Cell media was collected over a 3-day period beginning 2 days post-transfection and spun down in order to collect supernatant. Then, lentivirus was recovered from supernatant using an Amicon Ultra Centrifugal 50 kDa filter (MilliporeSigma) and transduced into HeLa cells along with 10 µg/ml polybrene (Santa Cruz Biotechnology, Dallas, TX). Cells were transduced sequentially with PKAR-FKBP-FP followed by Lyn-FRB. Each time, a selection procedure was completed, using 10 µg/ml puromycin or blasticidin, respectively, followed by generation of clonal lines by limiting dilution. Following selection, stable cells were cultured in DMEM media containing 1 µg/ml puromycin and blasticidin to maintain expression of the PKA-R translocation system.

## Microfluidic device fabrication

Multilayer microfluidic devices were fabricated from polydimethylsiloxane (PDMS) via replica molding from custom silicon masters, as described in Appendix 2. Devices were cleaned with 2% Alconox and 70% ethanol and then thermally bonded to #1.5 glass coverslips (Corning) by a 24 hr bake at 85°C. See Appendix 2 for further details.

## Microfluidic device setup

Microfluidic channels were coated with 10 µg/ml fibronectin by incubation for 1 hr at room temperature. Cells were suspended in imaging medium (DMEM without phenol red, 10% FBS, 1% penicillin/streptomycin) and seeded into the device as described in Appendix 2. Following overnight incubation at 37°C, a microfluidic valve was depressed to fluidically separate the cells from all future inputs and imaging medium containing rapamycin (or an equivalent volume of DMSO) was added to one or both sides of the device. A gradient from one side of the microchannels to the other was established by creating a height-driven pressure differential between two input and one output ports. Following the start of the imaging, the microfluidic valve was gradually released to expose cells to the gradient.

## Imaging

Widefield imaging was performed on a Zeiss Axiovert 200 M epifluorescence microscope with motorized stage (Prior, Rockland, MA) and live cell incubation chamber with humidity and temperature control (PeCon, Erbach, Germany) set to 37°C and 5% $CO_2$. Cells were imaged using a 40×, 1.3 numerical aperture oil immersion objective (Zeiss, Oberkochen, Germany) and Cascade II:1024 EMCCD camera (Teledyne Photometrics, Tucson, AZ). Microscope automation was controlled with Slidebook 6.0 software (Intelligent Imaging Innovations, Denver, CO). PKA-R was imaged using YFP excitation and emission filters for HeLa PFY cells or mCherry excitation and emission filters for HeLa PFM cells. For nuclear tracking experiments, cells were transfected with H2β-mCerulean, which was detected using CFP excitation and emission filters. To better identify the cell boundary for translocation analysis, HeLa PFM cells were stained with Vybrant DiO Cell-Labeling Solution (ThermoFisher Scientific). All FRET images were obtained using a CFP excitation filter as well as CFP and YFP emission filters. Semrock filters and corresponding dichroics were used (IDEX Health & Science, LLC, Rochester, NY). For the migration experiments, a spectral 2D template autofocus algorithm was employed using the phase channel to correct for focus drift between time points.

Confocal imaging was performed on a Nikon TiE inverted microscope (Nikon, Tokyo, Japan) equipped with a Yokogawa CSU-W1 spinning disk with 50 µm disk pattern (Yokogawa Electric, Tokyo, Japan) and CFI Plan Fluor 40×, 1.3 numerical aperture oil immersion objective (Nikon). Images were captured using an Andor iXon Ultra888 EMCCD camera (Oxford Instruments, Abingdon, UK). The microscope was equipped with a stage top incubator (Okolab, Pozzuoli, NA, Italy) maintaining humidity and 5% $CO_2$.

## Live cell imaging

For dish experiments, 35 mm glass bottom dishes #1.5 (Matsunami, Osaka, Japan) were coated with 5–10 µg/ml fibronectin (MilliporeSigma) for 1 hr at room temperature, and cells were incubated on the coated surfaces overnight prior to imaging. For FRET analysis, cells were imaged in Hank's balanced salt solution (ThermoFisher Scientific) to reduce background. All other imaging experiments were completed in the normal cell culture medium (see above) without phenol red. For analysis of translocation, images were taken every minute in the YFP and/or mCherry channels. For FRET, images were taken every 2 min in the CFP and FRET channels. HeLa PFM cells were used for all FRET experiments to avoid spectral overlap with YFP/CFP FRET probes. All other experiments were performed using HeLa PFY cells, except for the initial co-localization experiment shown in *Figure 1*, which was performed using HeLa cells transiently transfected with PKAR-FKBP-FP and Lyn-FRB, as a proof of principle.

## FRET image analysis

All image analysis was performed using custom MATLAB codes. For FRET images, image registration was performed prior to analysis. Images were segmented by intensity thresholding in the YFP channel. The FRET ratio was calculated as (FRET-DF)/(CFP-DF), where FRET is the intensity of emission collected in the YFP channel following CFP excitation. Cells were excluded from the analysis if the expression of the FRET probe was below a threshold value (as determined by fluorescence intensity) that was consistent across experimental replicates.

## Cytoplasmic intensity tracking

To verify translocation of PKA-R and PKA-C to the PM, the cytoplasmic intensity of fluorescently tagged PKA-R or PKA-C was tracked over time using images captured by spinning disk confocal microscopy. Post-imaging analysis was performed by: (1) segmenting the cells using YFP fluorescence; (2) eroding away the outermost 20 pixels (7.78 µm) of the segmented cell, leaving only the fluorescence intensity values for the non-PM compartments; and (3) taking an average of the non-zero pixels at each time point in the mCherry channel. This average was tracked as a function of time before and after rapamycin stimulation. Cells were excluded from the analysis if the intensity in the fluorescence channel used for segmentation was too low to properly distinguish cell from background.

## Cell migration analysis by nuclear tracking

Prior to seeding into microfluidic devices, cells were transfected with H2β-mCerulean for nuclear visualization and tracking. Following experimentation, images were registered in the phase channel to prevent any miscalculations of nuclear position due to stage drift. Then, the nucleus was identified by intensity thresholding in the CFP channel, the centroid of each nuclear ROI was calculated, and its location was tracked over time. Since cells were confined to motion in one dimension and devices were aligned such that microchannels were parallel to the stage mount, only the y position of the centroid was required to capture the change in nuclear position with respect to time zero. This positive or negative change in nuclear position was graphically presented for both single cells and as an average of cells in a population.

## Immunoblotting

HeLa cells transiently or stably expressing our PKA-R translocation system as well as untransfected controls were washed with ice-cold PBS, lysed with RIPA lysis buffer and Halt Protease and Phosphatase Inhibitor Cocktail (ThermoFisher) according to the manufacturer's protocols, and collected for immunoblotting. Laemmli buffer was added to lysates before heating them at 95°C for 5 min. Samples were allowed to cool and then loaded into 10% Mini-PROTEAN TGX Stain-Free protein gels for electrophoresis (Bio-Rad, Hercules, CA).

Following gel electrophoresis, proteins were transferred to a nitrocellulose membrane. The membrane was blocked with 5% BSA in Tris-buffered saline with 0.1% Tween-20 (TBST) and incubated with primary antibodies overnight at 4°C. Following additional washes with TBST, the membrane was incubated with horseradish-peroxidase-coupled secondary antibody for 1 hr at room temperature, washed again with TBST, and incubated in ECL Western blotting substrate (Promega) for protein visualization on a ChemiDoc XRS System (Bio-Rad). The primary antibodies used in this study are purified mouse anti-PKA[C] at 1:1000 dilution (BD; 610981), anti-PKA RIIβ at 1:400 dilution (Santa Cruz Biotechnology, Dallas, TX; sc-376778), anti-PKA RIIα at 1:1000 dilution (Abcam, Cambridge, UK; ab38949), anti-PKA RI-α/β at 1:1000 dilution (Cell Signaling Technology, Danvers, MA; 3927), and anti-GAPDH at 1:1000 dilution as a loading control (Cell Signaling; 2118). HRP-linked anti-rabbit and HRP-linked anti-mouse (GE Healthcare, Chicago, IL) were used as secondary antibodies. Blots were stripped before probing for GAPDH or a second protein of interest. Immunoblot images were analyzed using Fiji software (*Schindelin et al., 2012*).

## Statistical analysis

Results of cell imaging experiments were presented either as single cell traces or as means with shaded regions indicating SEM. Numbers of biological and technical replicates are as indicated in each figure. A biological replicate is defined as a single cell in an imaging experiment. A technical replicate is defined as a single imaging experiment. Each technical replicate was performed in a different cell imaging dish or microfluidic chip. Statistical analysis was completed in Microsoft Excel. Comparisons between groups were conducted using a two-tailed Student's $t$-test. Differences were concluded to be significant for p values less than 0.05. For immunoblots, one-way ANOVA was performed in GraphPad Prism 9.3.1 followed by Tukey's multiple comparisons test. Differences between pairs were concluded to be significant if they had adjusted p values less than 0.05.

## Acknowledgements

We thank Dr Jin Zhang (UCSD) for the kind gift of the Lyn-AKAR4 FRET probe. We are grateful to Yegor Isakov for his assistance in creating the PKAR-FKBP-FP and PKAC-mCh transient expression plasmids. We are grateful to Drs Eric Chu and Dr Hao Chang for their assistance with microfluidic fabrication and molecular biology techniques.

## Additional information

### Funding

| Funder | Grant reference number | Author |
|---|---|---|
| National Cancer Institute | U54 CA209992 | Rebecca LaCroix<br>Benjamin Lin<br>Tae-Yun Kang<br>Andre Levchenko |

The funders had no role in study design, data collection and interpretation, or the decision to submit the work for publication.

### Author contributions

Rebecca LaCroix, Conceptualization, Data curation, Formal analysis, Investigation, Methodology, Project administration, Software, Validation, Visualization, Writing - original draft, Writing - review and editing; Benjamin Lin, Conceptualization, Methodology, Software, Writing - review and editing; Tae-Yun Kang, Formal analysis, Investigation; Andre Levchenko, Conceptualization, Funding acquisition, Methodology, Project administration, Resources, Supervision, Writing - original draft, Writing - review and editing

### Author ORCIDs

Andre Levchenko (iD) http://orcid.org/0000-0001-6262-1222

### Decision letter and Author response

Decision letter https://doi.org/10.7554/eLife.66869.sa1
Author response https://doi.org/10.7554/eLife.66869.sa2

## Additional files

### Supplementary files

• Transparent reporting form

### Data availability

All data generated or analyzed during this study are included in the manuscript and supporting files. Source data files have been provided for Figures 1-4 and Figure 1 - figure supplement 1.

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

## Appendix 1

## Mathematical model

Here, we develop a heuristic model that approximates the key properties of linker proteins, such as the regulatory PKA subunit analyzed in this study. The complexity of this regulation precludes a simple precise analytical model, but some of its features may emerge from convenient approximations, particularly at high linker protein levels. The key to this analysis is the general assumption of our study that only a fully occupied linker molecule (e.g., the regulatory subunits of PKA-binding four cAMP molecules) will be fully active. For any molecule $S$, capable of binding $N$ molecules of the species $A$, one can show (**Buchler et al., 2003**) that the probability of all the binding sites occupied (fully occupied [FO]) will be described by the following expression, under the assumptions that the binding affinity of $A$ to all binding sites is the same (with the association equilibrium constant $K_A$), and denoting $A$ as the concentration of unbound molecules of this species:

$$P_{Fo} = \frac{N(K_A A)^N}{1 + \sum_{n=1}^{N} n(K_A A)^n}$$

Since, experimentally, we varied the local concentration of molecules S rather than $A$, it is important to consider how the above expression may depend on variable $S$ under the assumption of constant $A$. First consider the extreme of concentration of $S$ much higher than both $K_D = 1/K_A$ and $A$. Further, consider binding sites of $S$ (whose concentration is denoted $BS$) as independent from each other. Then the simple mass action kinetics yields to following expression for the concentration of these binding sites occupied by $A$:

$$C = \frac{A_T . BS}{BS + K_D}$$

Here $A_T$ denotes the total concentration of $A$. Under the assumptions above, the unbound concentration of $A$ is then:

$$A = A_T \left(1 - \frac{A_T . BS}{BS + K_D}\right) \cong \frac{A_T K_D}{BS} = \frac{\delta}{N \cdot S}$$

In the formula above, we denoted $\delta = A_T K_D$ and took advantage of the fact that concentration of BS is assumed to be much higher than $K_D$, and that there are $N$ binding sites in a molecule of $S$. This enables us to change the variables in expression for $P_{FO}$ from $A$ to $S$:

$$P_{FO} = \frac{N(\frac{K_A \delta}{N \cdot S})^N}{1 + \sum_{n=1}^{N} n(\frac{K_A \delta}{N \cdot S})^n} = \frac{N}{\tilde{S}^N + \sum_{n=1}^{N} n\tilde{S}^{N-n}}$$

In the equation above, we denoted $\tilde{S} = K_A \delta / NS = A_T / NS$. Noting that the concentration of the fully occupied $S$ is the product of $P_{FO}$ and the concentration of $S$, we have:

$$S_{FO} = \frac{NS}{\tilde{S}^N + \sum_{n=1}^{N} n\tilde{S}^{N-n}} = \frac{A_T \tilde{S}}{\tilde{S}^N + \sum_{n=1}^{N} n\tilde{S}^{N-n}}$$

Interestingly, although the expression above was derived under the assumption that $S \gg A$, $K_D$, one can easily see that it also captures the expected <u>full</u> occupancy of $S$ by $A$, when $S \ll A$, $K_D$. We therefore felt justified in using the formula above as the heuristic expression for the overall dependence of functional complex concentration on the concentration of the linker molecule. For the specific example of $N = 2$, we therefore have the following expression for the fully occupied linker molecule $S$:

$$S_{FO}(2) = \frac{A_T \tilde{S}}{\tilde{S}^2 + \tilde{S} + 2}$$

And for the example of $N = 4$ (e.g., corresponding to four binding sites for cAMP on PKA holoenzyme, the expression yields):

$$S_{FO}(4) = \frac{A_T \tilde{S}}{\tilde{S}^4 + \tilde{S}^3 + 2\tilde{S}^2 + 3\tilde{S} + 4}$$

We used these more specific expressions to illustrate in the main text that the model can indeed capture the biphasic behavior of the PKA regulation.

*Appendix 1—figures 1 and 2* show, in particular, that a gradual increase in the local abundance of the linker molecule *S* over time (if occurring in the quasi-steady-state manner) can progressively increase and then decrease the concentration of the fully occupied linker molecule and thus the signaling output, assuming that the fully occupied molecular state is crucial for the maximal signaling amplitude. This result is consistent with the experimental data in *Figure 2* of the main text. In particular, the end point of the experiment may be both above and below the baseline signaling levels, depending on whether there was an initial local level of the occupied molecule *S* (panel A vs. panel B in *Appendix 1—figures 1 and 2*). This result suggests that the cellular populations analyzed in our experiments had different pre-existent PKA-R levels at the PM prior to addition of rapamycin and initiation of enforced PKA-R translocation.

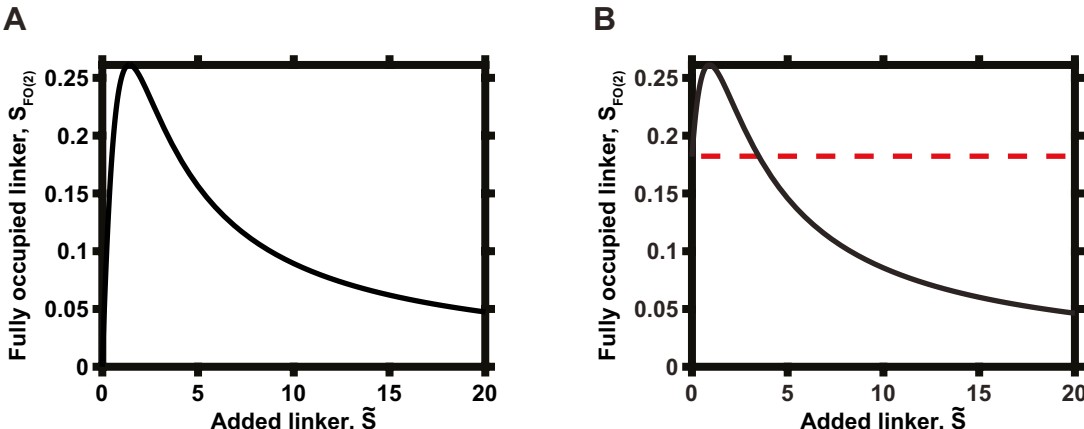

**Appendix 1—figure 1.** Overall dependence of functional complex concentration on the concentration of linker when $N = 2$. (**A**) Zero initial linker concentration. (**B**) Non-zero initial linker concentration, prior to addition of the indicated linker molecular concentrations. The initial concentration of the linker is indicated by dashed red line.

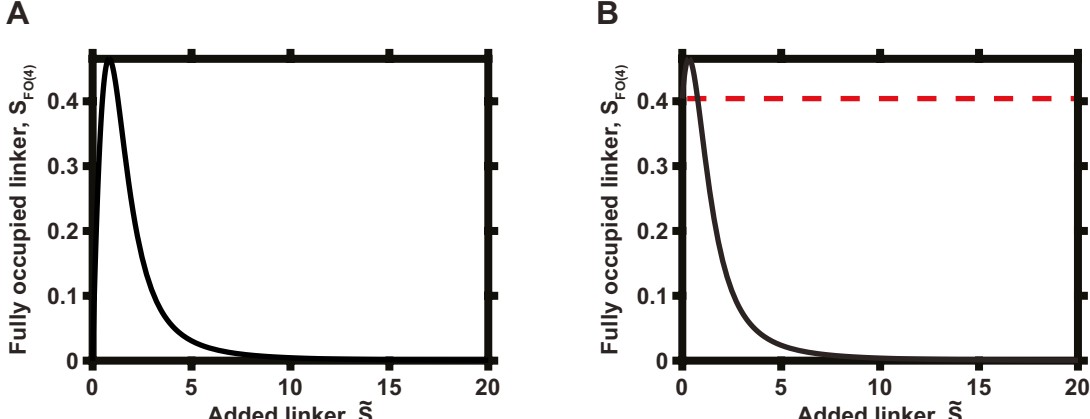

**Appendix 1—figure 2.** Overall dependence of functional complex concentration on the concentration of linker when $N = 4$. (**A**) Zero initial linker concentration. (**B**) Non-zero initial linker concentration. The initial concentration of the linker is indicated by dashed red line.

## Appendix 2

### Design of the microfluidic device for the analysis of the effect of PKA-R gradients on cell migration

To recapitulate the PKA gradients that have been seen in vitro, we needed a means of generating precise gradients of rapamycin. To do this, we utilized a diffusion-based microfluidic gradient generator (*Appendix 2—figure 1*) that is based on a previous design from the Levchenko lab (*Lin and Levchenko, 2015*; *Lin et al., 2015*). This device can generate soluble gradients that are stable over the time scale of our experiments (>6 hr). Briefly, a gradient is established within 6 μm tall parallel microchannels (gray) by passive diffusion from 130 μm tall perpendicular source and sink channels (light blue). The medium (and drug) is continually replenished by flow, which is driven by a hydrostatic pressure difference from two inlets (A and B or C and D) to a single outlet (O). Temporal control of experimental inputs is provided by a series of valves (dark blue) that, when depressed, can cut off flow to the microchannels. This two-layer design with valves was pioneered by the Quake group (*Unger et al., 2000*). Shear stress on cells seeded into the microchannels is minimized due to the large height difference between the source and sink and the microchannels, resulting in a much higher resistance in the microchannels and ensuring that the primary means of solute transport to the cells is diffusion. The microchannels are 16 μm wide, forcing the cells to assume a uniaxial phenotype. This reduces any analysis of cell migration to one dimension while providing a realistically confined environment. We increased the number of microchannels by over 20% from the original design (from 250 to 304) to increase the number of single cells that can be observed in parallel.

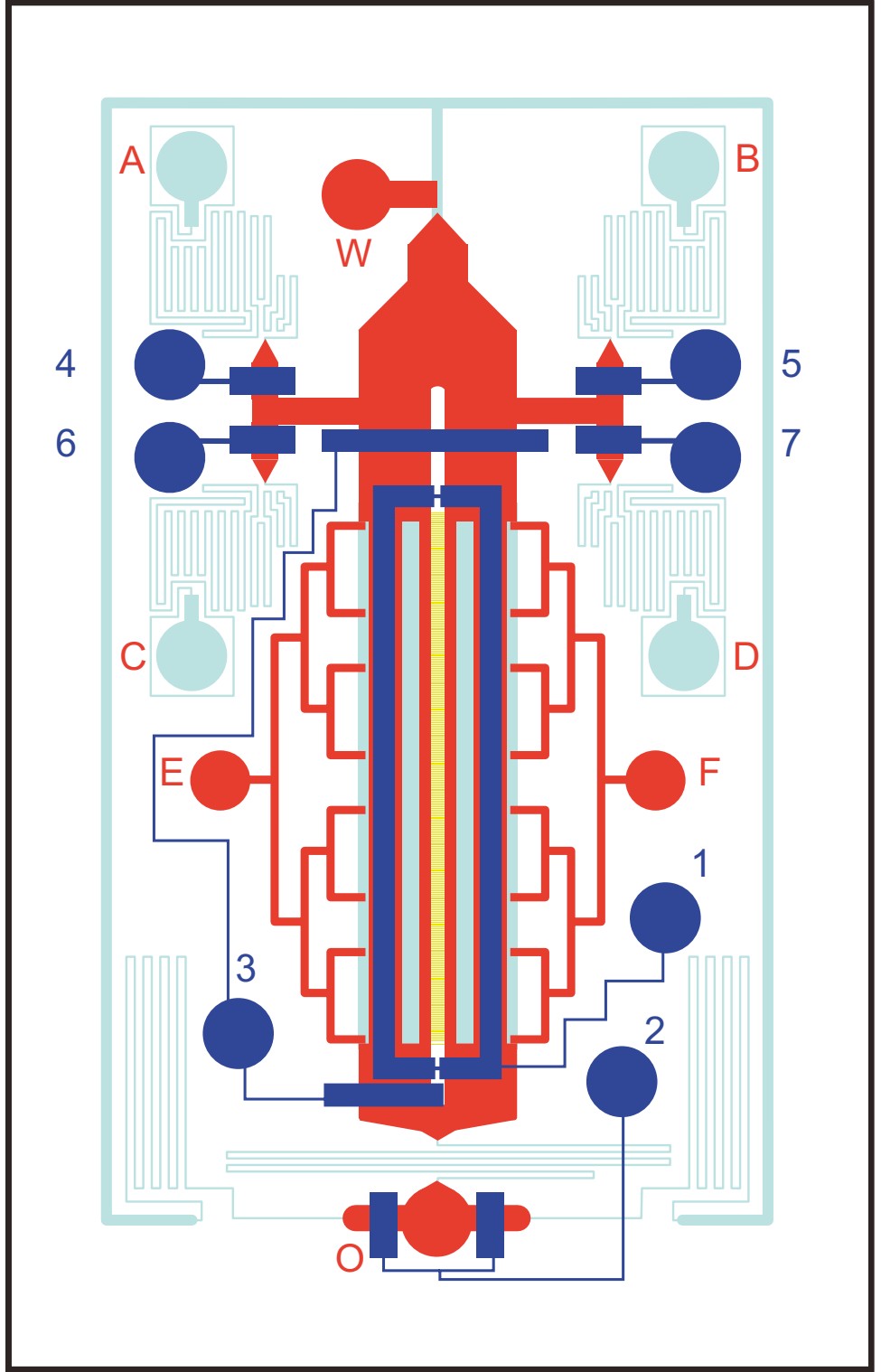

**Appendix 2—figure 1.** Microfluidic gradient chip. Cells are seeded into 16 µm wide × 6 µm tall microchannels (yellow) with gradient supplied by passive diffusion from adjacent 130 µm tall source and sink channels (light blue). Valves (dark blue) provide pressure to depress underlying 30 µm tall rounded fluidic channels (red). During experiment, medium (with or without drug) flows from inlet ports A/B or C/D to outlet port O.

## Microfluidic device fabrication

Microfluidic devices were fabricated from PDMS via replica molding from silicon masters. The silicon masters were fabricated in a clean room using a process known as photolithography in which photo-sensitive resists are spin-coated onto silicon wafers and regions corresponding to desired features are selectively cured with UV light (*Sia and Whitesides, 2003*). Photomasks for each master were designed using Adobe Freehand or Illustrator software and printed as transparencies.

Since our device consists of two layers of PDMS (a flow layer and an overlying valve, or control layer), two separate masters were prepared. For the control layer, SU-8 2025 (Microchem) was deposited onto a 3" silicon wafer and spin-coated according to manufacturer's protocol to achieve a desired feature height of 40 µm. Features were selectively exposed to UV light in a mask aligner with a constant energy of 1000 mJ/cm$^2$. Following exposure, the SU-8 was baked at 95°C for 1 hr followed by removal of uncured resist with SU-8 developer (Microchem) and a second bake at 200°C for 1 hr to reinforce the features.

For the second master, a similar protocol was followed. The flow layer includes features of three different heights: 6, 30, and 130 µm. Thus, the preparation of the master was conducted in three stages with alignment marks to register the features. SU-8 2005 (Microchem) was spin-coated onto a 3" silicon wafer to form the 6 µm level. Then, SPR-220–7 resist (Megposit) was spun onto the wafer at 1000 rpm for 30 s to form the 30 µm level Finally, SU-8 2025 (Microchem) was spun onto the wafer at 1500 rpm according to manufacturer's protocol to form the 130 µm level. In-between each spin-coating step, standard photolithography procedures were followed for UV exposure and baking of the channel regions as outlined above. For the SPR layer, a special hard bake protocol was followed to create rounded channels that could be depressed by pressurizing the overlying valves. Specifically, the wafer was baked for 5 hr while ramping the temperature at a rate of 180°C per hour to a final temperature of 200°C.

PDMS devices were fabricated by replica molding from the silicon masters. First, masters were functionalized with 1*H*,1*H*,2*H*,2*H*-perfluorooctyltrichlorosilane overnight to facilitate removal of PDMS from the mold. Then, PDMS (RTV615, Momentive) was prepared in a 20:1 ratio for the flow layer and 5:1 ratio for the control layer. Following initial degassing, PDMS was poured over the flow layer wafer and placed in a vacuum chamber for an additional 15 min to ensure that no air bubbles remained in the channel regions. The wafer was then spin-coated for 1 min at 390 rpm to achieve a uniform, thin layer of PDMS and then baked at 70°C for 15 min until partially cured. For the control layer, PDMS was poured over the silanized master and baked for 40 min at 70°C. Then, the valves were cut out, inlets were added using a 20-gauge luer stub, and valves were aligned on top of the flow layer. The two layers were then cross-linked to one another by baking at 70°C for at least 2 hr. Complete devices were cut from the silicon master and inlets added as before. Chips were cleaned with a combination of tape, 2% Alconox solution, and 70% ethanol prior to thermal bonding to #1.5 glass coverslips (Corning) at 85°C for 24 hr.

## Microfluidic device setup

Control valves 1–3 (*Appendix 2—figure 1*) were filled with DI water using 10 PSI pressure driven by a set of solenoid valves (the Lee Company). Then, the pressure to valve 1 was released, and valves 2 and 3 were pressurized to ~20 PSI to block flow from passing between the microchannel region and the rest of the fluidic network. A syringe containing 10 µg/ml fibronectin was inserted via syringe tubing into port E and elevated to force flow through the microchannels to ports F and O, which were then blocked.

Following 1 hr of fibronectin coating, the fibronectin was rinsed out with imaging medium (DMEM without phenol red, 10% FBS, 1% penicillin/streptomycin) through a syringe at port F. The rest of the fluidic network was filled through temporary release of valve 2 and insertion of syringes with imaging medium at either ports A and B or C and D. All other ports were blocked.

Next, a cell suspension was added as a droplet at port E, and cells were flowed into the microchannel region via negative pressure provided by lowering the syringe at port F. Once sufficient cells had been added and were lining the microchannels, valve 1 was pressurized to avoid disturbing the cells with future manipulations. The syringe at port F was then removed and a syringe with fresh imaging medium was added at port O. After blocking all ports, the valves were slowly released, and the device was transported to a 37°C incubator with 5% $CO_2$. Care was taken to ensure that all three syringes were kept at equal heights to prevent flow through the device. After

3 hr, the syringe in port O was lowered by 1 inch to maintain low velocity flow of fresh medium during overnight incubation.

Following overnight incubation, valve 3 was depressed to isolate cells from future manipulations. Then, one or both syringes at the top of the device were replaced with medium containing rapamycin and Alexa Fluor 594 dye to track the development of the gradient (ThermoFisher Scientific). The device was then transferred to the microscope, and a 2.5-inch pressure difference was established between the inlet and outlet ports to drive fluid flow and maintain the gradient. Following the start of imaging, valve 3 was slowly released to expose cells to the rapamycin gradient.

