## [Editor Report]

This is a very thorough and important study demonstrating quantitative control of signaling through changes in the abundance and localization of a regulatory kinase subunit. The authors use live imaging experiments in microfluidic devices to reveal nonmonotonic dependence of PKA activity on the level of its regulatory subunit and provide evidence that it translates into corresponding changes of cell polarization and cell migration. Moreover, they provide a mathematical model that explains the underlying mechanism.

---

## [Decision Letter]

**Decision letter after peer review:**

Thank you for submitting your article "Complex effects of kinase localization revealed by compartment-specific regulation of protein kinase A activity" for consideration by *eLife*. Your article has been reviewed by 3 peer reviewers, including Volker Dötsch as Reviewing Editor and Reviewer #3, and the evaluation has been overseen by Philip Cole as the Senior Editor. The following individual involved in review of your submission has agreed to reveal their identity: Stefan Knapp (Reviewer #2).

The reviewers have discussed their reviews with one another, and the Reviewing Editor has drafted this to help you prepare a revised submission. All reviewers agreed that the experiments reported interesting results on the regulation of PKA activity by local concentration. The main criticism focuses on the use of overexpression and the not reported relative concentrations of PKA and PKAR-IIβ.

Essential revisions:

1) Information on endogenous levels of PKA and of the regulatory subunit should be provided to show that at least overall levels are within the physiological range. A IHC comparison would be most useful as PKA concentration could be probed at different cellular locations in the presence and absence of PKA pathway stimulation.

2) More information on the use of overexpression or stable expressing cell lines is needed in the description of the experiments and figure legends.

3) A mathematical model that describes the quantitative behavior of the system would be very useful.*Reviewer #1:*

This is a very thorough and important study demonstrating quantitative control signaling through changes in the abundance and localization of a regulatory kinase subunit. The authors use live imaging experiments in microfluidic devices to reveal nonmonotonic dependence of PKA activity on the level of its regulatory subunit and provide evidence that it translates into corresponding changes of cell polarization and cell migration. The presentation is very clear and the results are convincing. Given the quantitative nature of the results, the authors could try to summarize their observations using a mathematical model. This would make the discussion even more convincing.

*Reviewer #2:*

The authors developed a FRET based sensor system that allows for the positioning of PKA at the plasma membrane in the absence of direct pathway stimulation. The system consists of a rapamycin-inducible dimerization domain which was achieved by linking the binding partner of rapamycin, FKBP, to fluorescent tagged PKA regulatory subunit PKAR-IIβ (construct PKAR-FKBP-FP) and a second fluorescent protein linked to the membrane-targeting sequence of Lyn kinase (Lyn-FRB). This assay system was used to shed light on the question of how localization affects PKA signalling outputs including complex phenotypic responses such a polarity and cell migration. The study also addresses the role of the regulatory subunit which on the one hand is a potent inhibitor of PKA activity but it also targets PKA to cellular locations, a prerequisite for PKA dependent signalling, thus functional also as a local "activator". The authors indeed found evidence of this paradoxical activating role of the regulatory subunit, which was transient in some cell types. An innovative microfluidic device controlled rapamycin gradients revealing that cell polarity can be tuned by the slope of PKA-R gradient.

The developed inducible FRET system provides interesting insights into the role of PKA location and the regulatory subunit. However, the system relies on ectopic expression only and it is not clear how the expression level compares with endogenous PKA levels. The main cell system (HeLa) is not the most obvious choice of cell lines for these studies as these cells are genetically highly compromised.

As the authors nicely show that PKA activity is highly dependent on PKAR-IIβ concentration, the relative levels of this regulatory subunit should be compared to levels in a PKA relevant cellular system. It is not clear if transient transfections or stable cell lines were used. It would have been helpful to add this information to the figure legends.

*Reviewer #3:*

LaCroix et al., investigate the catalytic activity of the protein kinase PKA in the context of the translocation of the regulatory subunit PKA-R to the plasma membrane, For this forced translocation they use the rapamycin / FBBP system with one component tethered to the regulatory subunit and the other to a plasma membrane anchor. They observe a rapid translocation of the regulatory unit to the plasma membrane as well as a translocation of the catalytic subunit of the kinase. Probing the activity of the kinase they observe a complex behavior with the activity first increasing and then decreasing. This behavior also depends on the rapamycin concentration used with lower concentrations – translating into a slower translocation of the regulatory subunit to the plasma membrane – resulting in a slower but sustained increase in catalytic activity. They also demonstrate the effect by measuring cell migration using microfluidic devices and rapamycin gradients.

The interpretation of the authors is that for the full activity a dimer of the catalytic subunit must be bound to the dimeric regulatory unit which then gets activated at the plasma membrane by binding to cyclic AMP. If the concentration of the regulatory unit at the membrane increases the catalytic subunits will progressively bind to so far un-complexed regulatory subunits which will – after an initial increase in activity – decrease the activity again. This interpretation is logical. A limitation, however, is that the authors do not provide measures of the (relative) cellular concentrations of the catalytic and the regulatory subunits. Since they use a lentiviral or transient transfection system it remains unclear if the observed effect is relevant at all under basal expression conditions or are created by the overexpression of one of the components.

---

## [Author Response]

Essential revisions:1) Information on endogenous levels of PKA and of the regulatory subunit should be provided to show that at least overall levels are within the physiological range. A IHC comparison would be most useful as PKA concentration could be probed at different cellular locations in the presence and absence of PKA pathway stimulation.

We are grateful for this comment, as it prompted us to explain more clearly the logic behind the use of the specific regulatory PKA subunit isoform (PKA-RIIβ) in our experiments. The key consideration for us was not to match the physiological levels of a regulatory subunit to the physiological levels in the cell, but rather to develop and use a method for control of the local abundance of this subunit in a specific cell compartment. This might mean that the local level of this subunit may be higher or lower than the physiological levels, to enable assessment of how these controlled variations may affect the PKA activity output. This consideration defined two specific choices we made in this study. First, we wanted to have the ability to assess the real-time local subunit abundance through imaging the labeled subunits, with minimal interference from the same unlabeled endogenous subunit isoform. Second, we attempted to control the local abundance of this subunit not through expression alone, but, more importantly, through the chemically induced recruitment to a specific compartment, in this case the plasma membrane. Thus, over the course of the experiment, the local abundance can acutely change from relatively low to high, revealing the effect of these changes on the activity of the enzyme.

The first choice above was enabled by our preliminary analysis, now explicitly shown in the new version of the manuscript, that revealed that of the 4 PKA-R isoforms (RIα/β, RIIα, and RIIβ) for a standard HeLa cell line, the expression of RIIβ was particularly low (the new Figure 1 —figure supplement 1). Further immunoblotting analysis revealed that HeLa cells with transient or stable overexpression of our modified PKA-RIIβ (PKAR-FKBP-FP), had a relatively lower level of overexpression of this isoform in transiently transfected cells and a higher level of overexpression in the stable cell lines (used for most experiments). While statistically insignificant, we did notice slightly lower expression levels of the RI and RIIα isoforms when RIIβ was overexpressed, suggesting possible downregulation of a compensatory nature. Notably, PKA-C levels were not affected by overexpression of PKA-RIIβ. These results justified the choice of PKA-RIIβ subunit for further analysis. Finally, and most importantly, in the preliminary characterization of this PKA-R analysis system, we found that, prior to stimulation, the labelled PKA-RIIβ was primarily localized in the cytosol, and thus away from the target compartment of the plasma membrane (Figure 1C). On the other hand, in the presence of chemical stimulation, the modified PKA-RIIβ translocated from the cytosol to the plasma membrane, with the concurrent changes in the membrane PKA activity explored in detail in the manuscript. Thus, chemically induced re-localization to a targeted compartment (our second experimental design choice) was robustly achieved in all cases.

We found no evidence of any phenotypic effects of the basal PKA-RIIβ overexpression in HeLa cells, whereas there were profound effects of its chemically induced plasma membrane translocation, suggesting that an increased abundance of this isoform in the cytosol had minimal consequences for cell behavior more generally, and for the control of cell shape polarization and migration, more specifically (likely due to limited cAMP abundance and thus basal PKA activity in the cytosol).

Overall, we would like to again reiterate that the purpose of the specific experimental design was to not only measure PKA activity per se, but to observe the consequences of the changes in this activity and the associated phenotypic responses, following controlled perturbations of the local PKAR concentration in a specific cellular compartment. We hope that this clarification, and the additional analysis now presented, will justify the experimental design we used and will illustrate the power of this technology in examining the function of various intracellular proteins.

2) More information on the use of overexpression or stable expressing cell lines is needed in the description of the experiments and figure legends.

More information on the use of transient overexpression vs. stable expressing lines has been added to the experimental methods and figure legends.

3) A mathematical model that describes the quantitative behavior of the system would be very useful.

A math supplement has been added and referenced in the text (Appendix 1). It represents a heuristic mathematical model that is consistent with our experimental observations, while providing an explicit mathematical underpinning of the proposed linker or scaffold function of PKA-R, previously discussed only as a phenomenological possibility. Both the supplement and the new portion of the main text discuss the modeling results in combination with the experimental data. We are thankful for this very helpful comment, as it (a) led us to develop an interesting new model of the process, and (b) substantially strengthened and clarified the mechanistic hypothesis of PKA-R function presented in the text.